DATA RELEASE

# First *De novo* whole genome sequencing and assembly of mutant *Dendrobium* hybrid cultivar 'Emma White'

Rubina Sherpa[1,2,†], Ramgopal Devadas[2,*,†], Penna Suprasanna[3], Sadashiv Narayan Bolbhat[1] and Tukaram Dayaram Nikam[4]

1  Department of Botany, Annasaheb Awate College, Manchar, Ambegoan 410503, Maharashtra, India
2  ICAR-National Research Centre on Orchids, Pakyong 737106, Sikkim, India
3  Nuclear Agriculture and Biotechnology Division, BARC, Mumbai 400085, Maharashtra, India
4  Savitribai Phule Pune University, Pune 411007, Maharashtra, India

## ABSTRACT

The *Dendrobium* hybrid cultivar 'Emma White' is an ornamental, successfully commercialised orchid. We used a gamma ray-induced early flowering mutant and the Illumina HiSeqX10 sequencing platform to generate the first draft *de novo* whole genome sequence and assembly. The draft sequence was 678,650,699 bp in length, comprising 447,500 contigs with an N50 of 1423 and 33.48% GC content. Comparing 95,529 predicted genes against the Uniprot database revealed 60,741 potential genes governing molecular functions, biological processes and cellular components. We identified 216,232 simple sequence repeats and 138,856 microsatellite markers. Chromosome-level genome assembly of *Dendrobium huoshanense* was used to RagTag-scaffold available contigs of the mutant, revealing a total length of 687,254,899 bp with an N50 of 2096. The longest final contiguous length was 18,000,059 bp from 30,571 bp. BUSCO genome completeness was 93.6%. This study is valuable for investigating the mechanisms of mutation, and developing *Dendrobium* hybrid cultivars using mutation breeding.

**Subjects**  Genetics and Genomics, Bioinformatics, Plant Genetics

**Submitted:**  05 February 2022

*  Corresponding author. E-mail: ramgopal.devadas@icar.gov.in

†  Contributed equally.

Preprint submitted at https://doi.org/10.1101/2022.06.25.497579

## DATA DESCRIPTION

### Background

The genus *Dendrobium* belongs to the tribe Podochileae and the subtribe Dendrobiinae [1]. There are about 1200 species in the genus *Dendrobium*, distributed throughout Southeast Asia and the Southwest Pacific islands. *Dendrobium* has a genome size (1C) of 0.75–5.85 pg [2] with a diploid chromosome number of 38 [3]. *Dendrobium* hybrids are orchids with high commercial value, and high medicinal demand and potential. Seventy percent of Dendrobiums are exported from Thailand, with a global value of US$63.6 billion [4]. They are the second best-selling potted flowering plants in the USA [5]. Scope for breeding novel Dendrobiums is limited owing to the narrow genetic makeup of hybrids from *Dendrobium phalaenopsis* [6], which is geographically native to Australia. There are also intersectional cross-incompatibility issues with transferring favourable genes [7]. Reverse genetics through target induced local lesions in genomics (TILLING) strategies could offer a rapid solution to trait improvement through mutation plant breeding [8].

*Dendrobium nobile*, known as the 'noble orchid', is the official state flower of Sikkim, India [9]. Its complete chloroplast genome was recently deciphered [10], and several functional genomics studies in *Dendrobium* have uncovered the biosynthetic pathways of alkaloids with medicinal uses [11, 12]. Given the large number of species in *Dendrobium*, DNA barcoding systems have been developed and tested as conservation and authentication tools [13, 14]. However, whole genome sequencing and assembly has been conducted in only four *Dendrobium* species of medicinal economic value [15, 16], and only using National Center for Biotechnology Information (NCBI) resources. This limits our current understanding of phylogenetic diversity among species and their relationships at the inter- and intraspecific level, and the subsequent use of this knowledge in crop improvement programmes.

## Context

So far, there have been no reports of a sequence assembly for *Dendrobium* hybrid cultivars (NCBI:txid136990) or mutants [17]. We have applied gamma radiation to induce mutations leading to new variability for orchid genetic improvement. We chose a popular and highly adaptable *Dendrobium* hybrid cultivar, 'Emma White', which is derived from a complex cross made through a series of hybridization events using five Dendrobium species: *Dendrobium phalaenopsis* (six times), *Dendrobium tokai* (once), *Dendrobium stratiotes* (once), *Dendrobium gouldii* (twice) and *Dendrobium lineale* (once) as parents in pedigree between 1938 and 2006. It was developed by T Orchids, Malaysia, and was registered with the Royal Horticultural Society in 2006 [18].

The *Dendrobium* 'Emma White' hybrid cultivar is a highly cross-compatible variety when used as the female parent in hybridization programmes [7]. It easily responds to *in vitro* studies compared with other hybrids [19], and was used as one parent to develop a new *Dendrobium* breeding line, NRCO-42, which is registered with the Indian Council of Agricultural Research (ICAR) National Bureau of Plant Genetic Resources, India (accession number INGR 10073) [20]. For the first time, here we present the draft genome sequence of a gamma-induced mutant of *Dendrobium* hybrid cultivar 'Emma White' (Figure 1). This will be a valuable resource to assist with genetic improvement through future TILLING strategies.

## METHODS

### Sampling and DNA preparation

Protocorm-like bodies (PLBs) of 'Emma White' hybrid plants were irradiated with gamma radiation at 10–40 Gy to induce random mutations at 32.54 Gy/min using $^{60}$Co gamma irradiator (Gamma Chamber 5000) at the Bhabha Atomic Research Centre, Mumbai, following standard protocols [21]. PLBs were cultured *in vitro* up to M1V5 generation and plantlets were raised from 10, 20 and 40 Gy. Subsequently, all surviving plantlets generated were moved to harden off, then were grown in polyhouse conditions for phenotypic evaluation. Early flowering mutants were identified among 10 Gy plants with several positive traits: plant height, pseudostem length, leaf number, leaf size and spikes during flowering, when there was no or delayed flowering, as in the case of control, 20 Gy and 40 Gy mutant plants.

Genomic DNA was isolated using the CTAB method [22] from 10 mg of fresh leaves of the first mutant plants during flowering. DNA sequencing libraries were prepared using a DNA

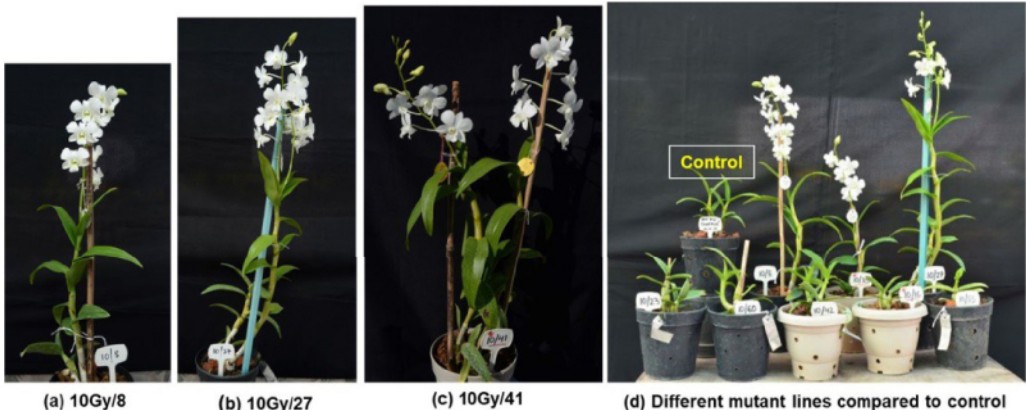

**Figure 1.** Early flowering mutant lines (10 Gy) of *Dendrobium* hybrid cultivar 'Emma White'. (a) 10 Gy/08 (b) 10 Gy/27 (c) 10 Gy/41 early flowering mutant lines. (d) Comparison of control plants with flowering mutant plants.

**Table 1.** BUSCO results of mutant *Dendrobium* hybrid cultivar and RagTag scaffolded assembly.

| BUSCO values | γ mutant assembly | | Scaffold assembly | |
|---|---|---|---|---|
| | **Number** | **(%)** | **Number** | **(%)** |
| Complete BUSCOs | 286 | 16.60 | 312 | 73.4 |
| Complete and single-copy BUSCOs | 249 | 15.43 | 306 | 72.0 |
| Complete and duplicated BUSCOs | 19 | 1.18 | 6 | 1.4 |
| Fragmented BUSCOs | 433 | 26.83 | 86 | 20.2 |
| Missing BUSCOs | 913 | 56.57 | 27 | 6.4 |
| Total BUSCO groups searched | 1,614 | 100 | 425 | 100 |

library preparation kit (NEB NextUltra) and validated for quality using Agilent Tapestation. DNA fragmentation was performed according to the manufacturer's instructions to produce fragments with an average length of 150 bp, followed by 5′ and 3′ adaptor ligation. Paired-end sequencing was performed using the Illumina HiSeqX10 platform (RRID:SCR_016385). Raw reads were used for *de novo* assembly using MaSuRCA (v.4.0.3; RRID:SCR_010691) with default parameters [23]. Adapter Removal (v.2) [24] was used to get rid of the adapters, low quality reads and bases. Assembly statistics were made using QUAST (v.5.2.2; RRID:SCR_001228) [25], and levels of conserved genes were generated using BUSCO (Benchmarking Universal Single Copy Orthologs; v.5.2.2; RRID:SCR_015008) [26]. The chromosome-based assembly for *Dendrobium huoshanense* [27, 28] was used for RagTag scaffolding using RagTag (v.2.1.0) [29] with default parameters. We also ran BUSCO on the scaffolded fasta file using the viridiplantae_odb10 lineage dataset to evaluate the assembled scaffold quality and make comparisons (Table 1).

Simple sequence repeats (SSRs) of each scaffold were identified using MISA (v.2.1; RRID:SCR_010765) script [30], and primer design was done on the predicted SSRs using primer3 (v.2.3.6) with default parameters [31]. MaSuRCA (v.4.0.3; RRID:SCR_010691) assembled contigs were used to run the gene prediction model using AUGUSTUS (RRID:SCR_008417) [32]. Predicted genes were compared against the Uniprot database (RRID:SCR_002380) [33] using BLASTX (RRID:SCR_001652) [34] with an *e*-value cut-off of $10^{-3}$ to identify potential genes governing different pathways. The best BLASTX hit based on

**Table 2.** Genome assembly statistics of the gamma-irradiated *Dendrobium* hybrid mutant and RagTag scaffolding.

| Assembly | γ mutant (*Dendrobium* hybrid) [35] | *Dendrobium catenatum* [36] | *Dendrobium huoshanense* [27] | RagTag scaffold of γ mutant (RagTag.Scaffold) |
|---|---|---|---|---|
| #contigs (≥0 bp) | 635,396 | 286,396 | 2256 | 549,354 |
| #contigs (≥1000 bp) | 213,573 | 29,592 | 2256 | 163,119 |
| #contigs (≥5000 bp) | 8,302 | 3,709 | 1279 | 5190 |
| #contigs (≥10,000 bp) | 625 | 2,523 | 907 | 370 |
| #contigs (≥25,000 bp) | 7 | 1,684 | 448 | 29 |
| #contigs (≥50,000 bp) | 0 | 1,401 | 145 | 20 |
| Total length (≥0 bp) | 678,650,699 | 1,104,259,548 | 1,284,285,095 | 687,254,899 |
| Total length (≥1000 bp) | 439,221,924 | 1,016,149,702 | 1,284,285,095 | 471,383,498 |
| Total length (≥5000 bp) | 57,139,593 | 973,286,060 | 1,282,134,848 | 184,746,937 |
| Total length (≥10,000 bp) | 7,915,569 | 964,906,549 | 1,279,530,669 | 154,097,582 |
| Total length (≥25,000 bp) | 194,559 | 952,168,543 | 1,272,192,375 | 149,748,116 |
| Total length (≥50,000 bp) | 0 | 942,392,114 | 1,262,665,926 | 149,476,984 |
| Number of contigs | 447,500 | 64,087 | 2256 | 369,938 |
| Largest contig (bp) | 30,571 | 33,291,853 | 100,197,051 | 18,000,059 |
| Total length (bp) | 604,787,319 | 1,040,039,458 | 1,284,285,095 | 616,868,961 |
| GC (%) | 33.48 | 34.61 | 35.73 | 33.49 |
| N50 | 1423 | 1,149,703 | 71,787,458 | 2096 |
| N75 | 949 | 434,049 | 52,753,504 | 1039 |
| L50 | 105,200 | 184 | 8 | 46,924 |
| L75 | 228,327 | 553 | 13 | 154,552 |
| #Ns per 100 Kbp | 0.00 | 4167.30 | 123.03 | 1394.82 |

query coverage, identity, similarity score and description of each gene was filtered out by the sequencing provider using custom python scripts and gene ontology was assigned.

## DATA VALIDATION AND QUALITY CONTROL

Paired-end sequencing using the Illumina HiSeqX10 sequencing platform generated 17× genome coverage with 79,792,942 reads (150 bp) for a 10-Gy/46 gamma-irradiated mutant line. On the basis of data from the NCBI Sequence Read Archive (SRA; accession number SRR16008784), this hybrid taxonomically matched 8.30% with its closest species *Dendrobium catenatum*, followed by *Phalaenopsis equestris* at 0.55%. The resulting genome assembly was 678,650,699 bp long, with a total of 635,396 contigs, with the longest being 30,571 bp and shortest being 300 bp, and a mean value of 1068 bp. The N50 value was 1423, GC content was 32.48%, and there were 447,500 contigs (Table 2).

RagTag scaffolding of the mutant assembly, based on the reference genome of *Dendrobium huoshanense*, contained a total length of 687,254,899 bp; an increase of 8,604,200 bp with an N50 value of 2096 (Table 2). The final largest contig length of the RagTag scaffolded assembly increased to 18,000,059 bp from 30,571 bp, with increased N's per 100 kilobase pairs (Kbp) to 1394.82 from 0. When compared against the Uniprot database using BLASTX with an *e*-value cut off of $10^{-3}$, the 96,529 genes predicted by MaSuRCA resulted in 60,741 potential genes governing different molecular functions, cellular components and biological processes. BLAST results were filtered based on a cut-off of qcov >60% and pi identity >70% to ensure confidence of annotations.

We also identified 216,232 SSRs and designed 138,856 microsatellite primers; these will be useful for generating polymorphic differences among progenies and putative gamma-irradiated mutant lines.

Most BLASTX hits showed affinity with *D. catenatum* based on functional annotation of genes (Figure 2). BUSCO (v.5.2.2) analysis revealed 913 (56.57%) single-copy orthologs that

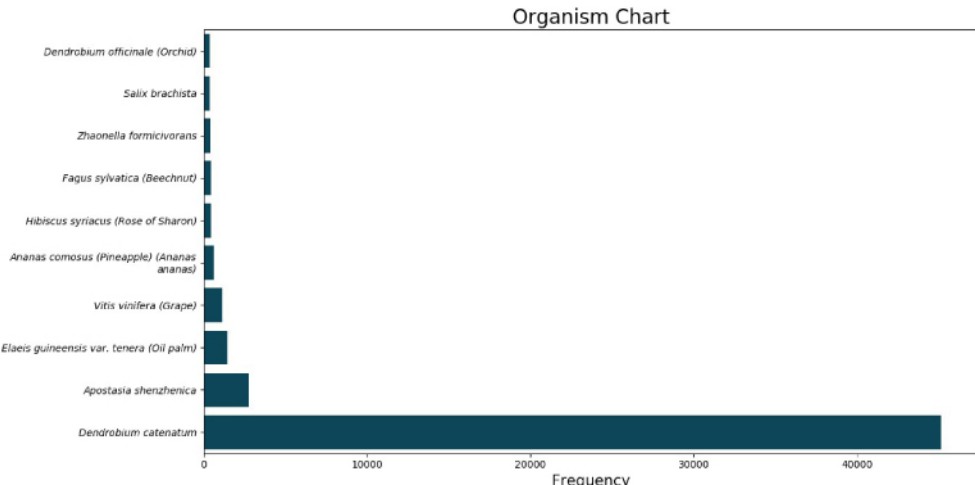

**Figure 2.** BLASTX hits based on functional predicted gene models for different organisms.

do not match with any databases; this indicates a possible effect of both the genomic background of the developed hybrid cultivar, and of the gamma radiation. The complex genome of the 'Emma White' hybrid *Dendrobium* cultivar is derived from five unique and unrelated species; it has been hybridized 11 times over a period of 68 years, with selection for targeted economic trait improvement (Table 1). Low BUSCO values may be attributed to its fragmented assembly. However, the presence of genomic material from several other species of same genus (otherwise contaminant species) in the hybrid cultivar may have resulted drastic changes in the missing BUSCO values [37–39]. NCBI taxonomical data for mutant *Dendrobium*, based on raw sequence data, also supports the view, since it has limited synteny with its closest relative, *Dendrobium catenatum*, at less than 9%.

In addition, multigenome hybrid cultivars are genetically heterogeneous and have an outcrossing nature, indicating higher compatibility. For example, the outcrossing species *Arabidopsis lyrata* had 32,670 predicted genes, even at 8.3× DNA coverage, compared with 27,025 genes in the selfing species *Arabidopsis thaliana* (125 megabase pairs [Mbp]), which diverged 10 million years ago [40] because of genomic loss and rearrangement. In a similar way, these novel *Dendrobium* hybrid cultivars have a distinct genome, because of introgression from other wild species chosen by plant breeders to create new genetic variations in a short space of time compared with evolutionary changes. It can also be attributed to deletions, mostly noncoding DNA and transposons, and the presence of a highly mutagenized background with severe developmental abnormalities; apart from presence of unclustered genes [41].

## REUSE POTENTIAL

The mutant *Dendrobium* hybrid sequencing and genome assembly presented here can be adopted as a primary reference genome, as well as complementing existing conventional *Dendrobium* species already in the public domain. Studies of induced mutants allow rapid discovery of new alleles at low cost using high throughput TILLING [42]. As evident from other crops [43, 44], this is especially true in vegetatively propagated *Dendrobium* hybrids to obtain high-density mutations using gamma radiation mutation breeding. These results



provide a baseline for further research on the molecular understanding of desired traits in mutant germplasm, and to develop genomic resources for orchid improvement.

## DATA AVAILABILITY

The *de novo* whole genome sequence of the gamma-irradiated mutant *Dendrobium* hybrid cultivar 'Emma White' (10 Gy) was deposited with the NCBI with SRA accession number SRR16008784 and Genbank assembly accession GCA_021234465.1 [35]. and also available in the public domain via BioProject ID PRJNA763052. Additional data is available in the *GigaScience* GigaDB repository [45].

## DECLARATIONS
## LIST OF ABBREVIATIONS

bp: base pair; BUSCO: Benchmarking Universal Single Copy Orthologs; ICAR: Indian Council of Agricultural Research; Kbp: kilobase pair; Mbp: megabase pair; NCBI: National Center for Biotechnology Information; SRA: Sequence Read Archive; SSR: single sequence repeat; TILLING: target-induced local lesions in genomics.

## ETHICAL APPROVAL

Not applicable.

## CONSENT FOR PUBLICATION

Not applicable.

## COMPETING INTERESTS

The authors declare that they have no competing interests.

## FUNDING

RD is funded by a research grant from the Board of Research in Nuclear Sciences, Government of India (Ref: 35/14/22/2016-BRNS/35061).

## AUTHORS' CONTRIBUTIONS

RS conducted the gamma radiation experiment, mutant plant development and genome analysis. RD prepared the concept and research formulation of the project and wrote the manuscript. PS assisted with radiation facilities and edited the manuscript. SNB and TKN supervised the work and interpreted the data. All authors read and approved the final version of the manuscript.

## ACKNOWLEDGEMENTS

The authors would like to acknowledge the Nuclear Agriculture and Biotechnology Division of the Babha Atomic Research Centre, Government of India, for providing radiation facilities. The authors are thankful to the Director of Indian Centre for Agricultural Research–National Research Centre on Orchids for providing institutional support. We also thank M/s AgriGenome Labs Private Limited, Kerala, India for the sequencing work.

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
