## [Reviewer Report]

Comments on revised manuscriptThank you to the authors for addressing the previous comments on the manuscript. The authors have followed up on the suggestion scaffold the genome by using the published Dendrobium huoshanense genome to scaffold their draft genome using RagTag. This is an appropriate tool to use and has improved the contiguity of the draft assembly which is good to see. In the methods, the version of RagTag is missing, as are the parameters used to run the program. Please also provide specification on the specific RagTag utilities used (correct, scaffold, patch and/or merge). The authors have added genome statistics for two other orchids and the scaffolded assembly in Table 1, however, have not added BUSCO results for their scaffolded assembly in Table 2. Also, can the authors provide a comment on if the low BUSCO values may be related to the fragmented assembly as brought up in the previous round of review? It will be interesting to see if BUSCO has improved with the scaffolding. BUSCO results for the other two species, D. catenatum and D. huoshanense, would also be a good point of comparison and this is relatively simple and quick to add. The authors could consider concatenating Table 1 and 2 in this case.

The draft assembly has improved, and the authors should report numbers on the final version of the assembly presented in the paper (i.e. the scaffolded assembly) in terms of the analysis they have run. In the results and discussion section, it appears some of the statistics (e.g. 96,529 genes, 216,232 SSRs) still refer to the first draft assembly.

The authors have clarified that raw reads were used as input into MaSuRCA (line 111) and have now included the necessary detail for the input and parameterisation of the program.

Line 157-159: “Taxonomical analysis of mutant Dendrobium at raw sequence data also revealed limited synteny with its closest Dendrobium catenatum species at below 9% and genetically heterogeneous with outcrossing nature”. Details of how this analysis was done is missing from the methods. It may be more appropriate to perform synteny analysis at the genome level and compare the published D. catenatum genome with the scaffolded Dendrobium hybrid genome.

---

## [Reviewer Report]

Comments on revised manuscriptThank you to the authors for addressing the previous comments on the manuscript. I generally find the revisions satisfactory, although have some follow up comments.

The addition of details on the genetic origin of the Dendrobium ‘Emma White’ hybrid cultivar and requested details on bioinformatic tool versions/parameters have strengthened the manuscript.

The authors have not followed up on the suggestion to improve the genome via scaffolding, but provide an explanation that existing chromosome-level assemblies/sequencing data of Dendrobium species are not suitable as they are not related to the hybrid cultivar the authors studied, implying that they are highly diverged and scaffolding would not meaningfully improve the genome. Given this information, I think the Dendrobium ‘Emma White’ hybrid cultivar genome can still be useful for orchid breeding efforts despite low contiguity and completeness. However, I do not agree with the author’s point of, “Third, we used low coverage genome analysis with short reads of gamma mutant Dendrobium hybrid cultivar, as it was the first case study and obtained SRA, genome assembly and TSA accessions from NCBI. The genome assemblies of Dendrobium species from earlier studies used both long reads and short reads in their study. Construction of scaffolding from such database species using our contigs may be skewed and shall give unreliable data based on above points mentioned. Hence, I opinioned that suggestion given by Reviwer 2 on scaffolding suggestion may not be correct point.” Even if different types of sequencing technologies were used in comparison to Emma White genome, the availability of a contiguous closely related reference genome would still be useful for reference-guided scaffolding of the draft genome and well as comparative analyses.

Lines 107-109: Reorder sentence to make the order of the steps clear i.e. adapter removal and quality filtering before assembly with MaSuRCA. Also, on the MaSuRCA GitHub (https://github.com/alekseyzimin/masurca), it says “Avoid using third party tools to pre-process the Illumina data before providing it to MaSuRCA, unless you are absolutely sure you know exactly what the preprocessing tool does. Do not do any trimming, cleaning or error correction. This will likely deteriorate the assembly.” Did the authors find that the pre-processing meaningfully improved the quality of the assembly, compared to if the raw reads were input straight into the assembler? Please justify the preprocessing of reads.

Suggest to reword lines 137-139 “BUSCO version 5.2.2 analysis reveals 913 (56.57%) single-copy orthologs doesn’t match with any data bases indicates the impact from evolutionary development of hybrid cultivars and influence of gamma radiation. It is because, the genome of ‘Emma White’ hybrid cultivar of Dendrobium derived from five unique different species is complex genome and continuously hybridized repeatedly 11 times over a period of 68 years with selection process for economic trait improvement” to make the explanation clearer and also to include the number and/or percentage of complete BUSCOs. This was flagged in the previous comments, but not fully resolved and would benefit from revisiting the interpretation of BUSCO results. There are a large number of missing BUSCOs in your assembly, likely related to low contiguity (as well as radiation which is mentioned). Can you discuss if/how this may be a limitation for using this genome in further studies? You suggest that the BUSCOs are not found in the assembly due to many rounds of trait selection and radiation. It is possible that some of the BUSCOs are indeed missing from the particular plant sequenced, but how can you be certain that this is due to the breeding history and radiation applied as implied in the text, and not low genome contiguity? Some papers which characterised gamma irradiation-induced mutations in plants (DOIs: 10.1093/jrr/rraa059, 10.1186/s12864-019-6182-3, 10.1534/g3.119.400555) indicate that it is unlikely as many as 913 BUSCO genes have been affected. Even with stronger doses of radiation than used on the orchid, the number of mutations/genes affected is much lower.

The genus name needs to be consistently italicised throughout the manuscript.

---

## [Reviewer Report]

Reviewer name and names of any other individual's who aided in reviewer Linzhou LiDo you understand and agree to our policy of having open and named reviews, and having your review included with the published papers. (If no, please inform the editor that you cannot review this manuscript.)YesIs the language of sufficient quality?YesPlease add additional comments on language quality to clarify if needed
Are all data available and do they match the descriptions in the paper? YesAdditional CommentsAre the data and metadata consistent with relevant minimum information or reporting standards? See GigaDB checklists for examples <a href="http://gigadb.org/site/guide" target="_blank">http://gigadb.org/site/guide</a>NoAdditional CommentsGeographic location (country and/or sea,region, latitude and longitude) is missing, as well as environmental context. Is the data acquisition clear, complete and methodologically sound?YesAdditional CommentsIs there sufficient detail in the methods and data-processing steps to allow reproduction?YesAdditional CommentsIs there sufficient data validation and statistical analyses of data quality? NoAdditional CommentsThe genome size and gene number of Dendrobium hybrid cultivar ‘Emma White’ differ greatly from the published Dendrobium genomes (e.g. Zhang et. al Scientific Reports 2016, Zhang et. al Horticulture Research 2021, Han et. al Genome Biology and Evolution 2020...). Specifically, the authors assembled a smaller genome and predicted a larger number of genes compared with the previous study. Therefore, I strongly suspect that the assembled genome is incomplete and fragmented, resulting in more fragmental genes. Is the validation suitable for this type of data?NoAdditional CommentsThere's not enough raw data (~24Gb) to assemble a 600Mb (or ~1.2Gb from the previous study) genome. I highly recommend the authors get more raw data and do a genome survey.Is there sufficient information for others to reuse this dataset or integrate it with other data?YesAdditional CommentsAny Additional Overall Comments to the AuthorThe Complete BUSCOs only account for 16.6% which is quite low. The authors explain that the large loss of BUSCOs is due to the fact that the mutant genome has a lot of specific sequences, but these genes are very conserved in plants and should not be easily mutated.RecommendationReject (Unsound or Unusuable)

---

## [Reviewer Report]

Upload additional filesDRR-202202-01/form/Review_SC_9676_050222_Gigabyte_Gamma WGS datanote (1).docxReviewer name and names of any other individual's who aided in reviewer Stephanie ChenDo you understand and agree to our policy of having open and named reviews, and having your review included with the published papers. (If no, please inform the editor that you cannot review this manuscript.)YesIs the language of sufficient quality?NoPlease add additional comments on language quality to clarify if needed
Most of the manuscript is written in a sufficient quality, but there are certain parts that require revision to improve readability. Please see detailed comments on the Word document.Are all data available and do they match the descriptions in the paper? NoAdditional CommentsThe SRA link is coming up as a permission error, but I assume it will be released once the paper is available. There is no information on where to access the annotation file.Are the data and metadata consistent with relevant minimum information or reporting standards? See GigaDB checklists for examples <a href="http://gigadb.org/site/guide" target="_blank">http://gigadb.org/site/guide</a>YesAdditional CommentsIs the data acquisition clear, complete and methodologically sound?NoAdditional CommentsThe contiguity (635,396 contigs, N50 of 1,620 bp) and completeness (16.60 %) of the genome is quite low and this may limit its downstream uses. It would be good to incorporate some long-reads or increased sequencing coverage to improve your genome. There are a number of chromosome-level Dendrobium genomes that are available (e.g. D. chrysotoxum and D. huoshanense) and scaffolding off these may be attempted to improve the assembly. Scaffolding from existing assemblies may be a good option if generating more sequencing reads is not feasible.Is there sufficient detail in the methods and data-processing steps to allow reproduction?NoAdditional CommentsSome details on the DNA extraction and library preparation steps are missing. In the methods section, there are also missing details for multiple programs in terms of the version and parameters (e.g. BUSCO version and database used, QUAST version, AUGUSTUS version, details on adapter removal and trimming). It is mentioned 'similarity score and description of each gene was filtered out using in-house pipeline'. The script and details of the pipeline are not provided; please add a reference or details in the manuscript e.g. link to GitHub repository.Is there sufficient data validation and statistical analyses of data quality? NoAdditional CommentsThe reporting and interpretation of BUSCO results ('BUSCO version 5.2.2 analysis reveals 913 (56.57%) single-copy orthologs doesn’t match with any data bases indicates the unique and possible uncharacterized sequences in mutant genome of Dendrobium hybrid cultivar') needs to be revisited. There needs to be additional validation of the gene annotation (e.g. BUSCO, comparison with existing Dendrobium annotations) and also some validation of the genome size (e.g. GenomeScope and comparison with reported flow cytometry measures).Is the validation suitable for this type of data?YesAdditional CommentsThe type of validation in the manuscript (BUSCO) is suitable to assess genome completeness, but reporting and discussion of the results needs to be revised. Some additional validation is also needed (see box above).Is there sufficient information for others to reuse this dataset or integrate it with other data?YesAdditional CommentsAny Additional Overall Comments to the AuthorIn this manuscript, the authors provide a draft genome of a gamma-ray induced mutant of a Dendrobium hybrid cultivar using Illumina sequencing that will assist with future breeding efforts and studies. However, I am not convinced of the genome's usefulness in its current form. There are some methods that need to be described in more detail to be reproducible. Revisions will also help improve the readability of the manuscript. As page and line numbers are not provided on the manuscript, please find additional comments directly added to manuscript file attached.RecommendationMajor Revision